# Mutation in glutamate transporter homologue GltTk provides insights into pathologic mechanism of episodic ataxia 6

Emanuela Colucci[1], Zaid R. Anshari [1], Miyer F. Patiño-Ruiz[1], Mariia Nemchinova[1], Jacob Whittaker [1], Dirk J. Slotboom [1] ✉ & Albert Guskov [1] ✉

Episodic ataxias (EAs) are rare neurological conditions affecting the nervous system and typically leading to motor impairment. EA6 is linked to the mutation of a highly conserved proline into an arginine in the glutamate transporter EAAT1. In vitro studies showed that this mutation leads to a reduction in the substrates transport and an increase in the anion conductance. It was hypothesised that the structural basis of these opposed functional effects might be the straightening of transmembrane helix 5, which is kinked in the wild-type protein. In this study, we present the functional and structural implications of the mutation P208R in the archaeal homologue of glutamate transporters $Glt_{Tk}$. We show that also in $Glt_{Tk}$ the P208R mutation leads to reduced aspartate transport activity and increased anion conductance, however a cryo-EM structure reveals that the kink is preserved. The arginine side chain of the mutant points towards the lipidic environment, where it may engage in interactions with the phospholipids, thereby potentially interfering with the transport cycle and contributing to stabilisation of an anion conducting state.

In the brain, glutamate and chloride are directly involved in the regulation of signal transmission at the synaptic level. While glutamate has the primary role of a neurotransmitter by stimulating ionotropic and metabotropic receptors after its release in the synaptic cleft[1,2], chloride has a more secondary role—securing a hyperpolarisation of the neurons and preventing the immediate reactivation of the signal. Excitatory amino acid transporters (EAATs) are secondary active transporters which play a key role in the maintenance of the homeostasis in synapses since they have evolved a double function as both transporters of L-glutamate and channels for chloride[3–6]. EAATs restrict the neurotransmitter levels within the synapses to avoid excessive and potentially neurotoxic concentrations of L-glutamate. Functional and structural characteristics of human glutamate transporters (EAATs 1-5), as well as their archaeal homologues (aspartate transporters $Glt_{Ph}$ and $Glt_{Tk}$), are well documented[2,5,7–9]. Briefly, all members of the glutamate transporters family (SLC1A) consist of three protomers subdivided into a rigid scaffold domain and a dynamic transport domain. The scaffold domain connects each independent protomer to the other and for this reason, it is also called the trimerization domain (Fig. 1a); the transport domain moves along the scaffold with a movement defined as a one gate elevator mechanism (reviewed in ref. [10]). The elevator movement is gated by the hairpin loop 2 (HP2) for the transport of glutamate (or aspartate in the case of archaeal proteins). Crucially, the formation of the binding site for the amino acid and its transport is strictly dependent on the binding of sodium ions[11–16]. During the turnover, when a transport domain slides along the scaffold domain, a transient aqueous channel is formed, which creates thermodynamically uncoupled anion conductance[17–22].

Despite the conserved structural and mechanistic features, there is functional variety among the members of the family. While EAATs 1-3 exhibit a high glutamate transport rate associated with low chloride conductance, EAATs 4 and 5 create a large anion current activated by a

---

[1]Groningen Institute for Biomolecular Sciences and Biotechnology, University of Groningen, 9747AG Groningen, the Netherlands.
✉e-mail: d.j.slotboom@rug.nl; a.guskov@rug.nl

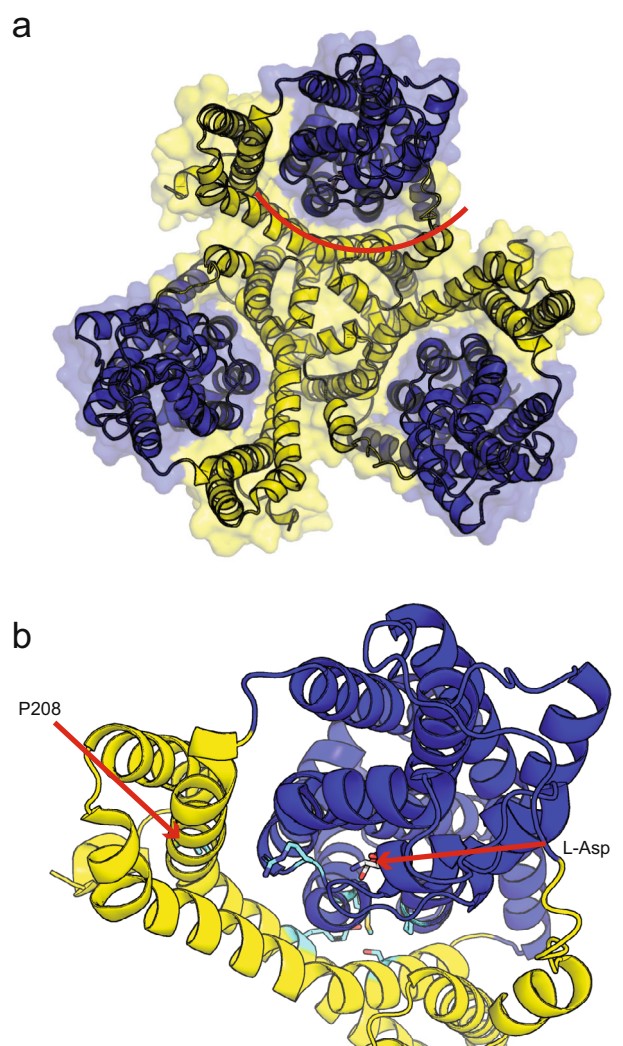

**Fig. 1 | Trimeric assembly of Glt$_{Tk}$ and conservation of the chloride channel. a** The general architecture of Glt$_{Tk}$ comprised of trimerization domains (yellow) and the transport domains (blue), view from the extracellular side of the membrane. The red line indicates the interface (in one of the protomers), where the anion-conducting channel is formed during the elevator movement of the transport domain. **b** Zoom-in onto one of the protomers (same orientation); residues involved in the formation of the pore are shown in cyan sticks, positions of the substrate L-Asp and P208 (homologous to P290 in hEAAT1) are indicated with the red arrows.

low rate of glutamate transport[23–31]. Furthermore, the family can be subdivided into the glial types (EAAT1 and EAAT2), which are capable of creating anion conductance in a substrate- and Na$^+$-independent manner and the neuronal types (EAAT3 and EAAT4) in which the anion conductance is strictly coupled to substrate and Na$^+$ binding and transport[32].

Despite the well-established double role of the glutamate transporters acting as amino acid transporters and chloride channels[3–6,18], structural insight into the chloride conducting state is limited. Evidence of a chloride channel in the archaeal homologues is based on the work involving Glt$_{Ph}$, where chloride-flux assays, fluorescence quenching experiments conducted on the tryptophan mutants, and molecular dynamic simulations led to the prediction of the channel formation between the trimerization and the transport domain distributed between the hairpin loop 1 (HP1), TM2, TM5 and TM7[21,33,34]

(Fig. 1b shows the positions of these conserved residues in our archaeal model, Glt$_{Tk}$). Cryo-EM structures providing insight in the chloride channel activity, are derived from the cross-linked structures of Glt$_{Ph}$ which mimic the intermediate state between the outward and the inward facing state, called chloride conducting state (ClCS) and which was used to determine the conduction pathway and calculate the free energy associated with its formation[22].

Interestingly, the single point mutation P290R in hEAAT1 converts this high-rate glutamate transporter associated with small anion conductance into a low-rate transporter that exhibits high chloride conductance[35,36]. In humans, this mutation is related to the rare neurological disease called episodic ataxia type 6 (EA6), described with episodes of hemiplegia and seizures accompanied by a progressive degeneration of the cerebellum[37,38]. The pathology that characterises EA6 is the consequence of the change in the glial chloride homeostasis. This is governed by the increase of the anionic channel activity, rather than a decrease of glutamate transport[39–42]. A few attempts have been made to explain this change at the molecular level. Voltage fluorimetry and voltage clamp experiments show that in the human proteins EAAT1 and EAAT3, the mutation of proline into arginine is associated with a reduction of glutamate uptake and an increase of anion current when compared with the wild type[35–37,43].

The highly conserved proline creates a kink in transmembrane helix 5, which is part of the trimerization domain (Fig. 1b). It has been suggested that the substitution of proline with arginine could abolish the kink and lead to a rearrangement in the protomer that could deform the binding sites for sodium and glutamate[36]. Furthermore, it has been hypothesised that the side chain of arginine at this position could form an electrostatic interaction with the phosphate head group of the lipids in the membranes[44]. Such an interaction would then be responsible for the formation of a wider chloride channel during the elevator movement. However, these hypotheses have not been substantiated experimentally.

In this work, we use the structurally and mechanistically well characterised homologue of human glutamate transporters Glt$_{Tk}$, and analyse the conformational and functional implications of mutating the essentially conserved proline at position 208 (P290 in hEAAT1, P259 in hEAAT3) into arginine (Fig. 1b). The presence of a substrate-gated anion channel in Glt$_{Tk}$ is demonstrated by solid-supported membrane electrophysiology (SSM). Combined with Isothermal Titration Calorimetry (ITC) and $^{14}$[C]-L-Asp transport assays the data show that the binding affinity for L-aspartate remains unchanged, the transport rate of L-aspartate is reduced by 2.5-fold in the mutant compared to the wild-type, and the anionic conductance is increased by 3-fold. A cryo-EM structure of the mutant protein embedded in lipid nanodisc surprisingly reveals no major rearrangement of the protein architecture, with the kink in TM5 preserved, and show that the arginine side chain is directed towards the lipidic environment. Molecular dynamics simulations support the formation of a stable salt bridge between the arginine and a phospholipid headgroup. Taken together, the opposite effects of this mutation on substrate uptake and anion conductance might be caused by the impediment in the transport cycle and/or by the widening of the anionic channel.

## Results
### Anion conductance in Glt$_{Tk}$ is associated with Na$^+$:L-Asp symport activity
To test if the chloride channel activity is conserved in Glt$_{Tk}$, we used solid-supported membrane electrophysiology (SSM) (Supplementary Fig. 1). All measurements were performed using double solution exchange experiments, which allow detection of electrogenic transport activity in proteoliposomes induced by a jump in the concentration of a substrate in the presence of a gradient of a second substrate[45]. The same type of experiment was done previously to gain insight into the transport mechanism of the murine homologue of EAAT3, mEAAC1[46].

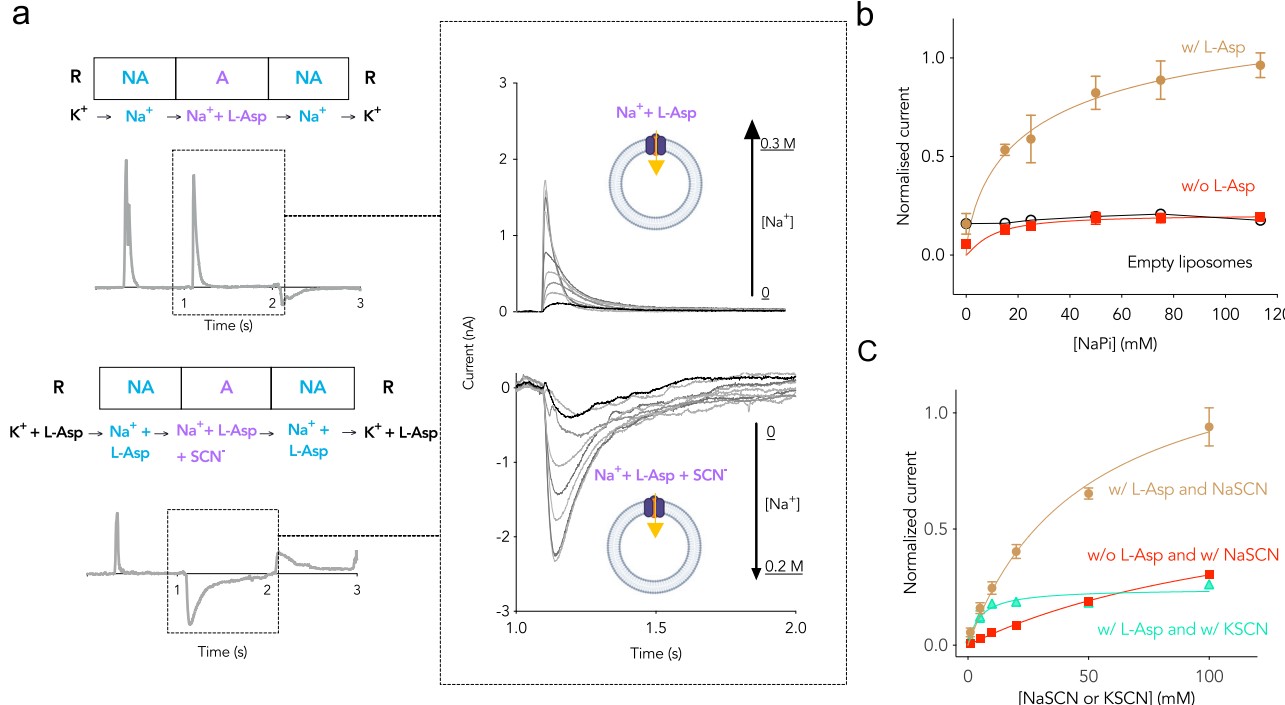

**Fig. 2 | Electrogenic transport activity and substrate dependence of the anion conductance in wild type Glt$_{Tk}$. a** Top, Left: flow scheme of current amplitude measurement upon L-Asp jump. Top Middle: Progressive increase of the current amplitude upon L-Asp jump in the presence of increasing concentrations of Na$^+$ in the activating buffer. **a** Bottom, Left: flow scheme of current amplitude measurement upon SCN$^-$ jump. Progressive increase of the negative current amplitude upon SCN$^-$ jump in the presence of increasing concentrations of Na$^+$ in the activating buffer. Details of the buffer exchanges are reported in the main text and the full traces are shown in Supplementary Fig. 3) (**b**) Na$^+$ dependence of the SCN$^-$ currents in the presence or absence of L-Asp measured on wild type Glt$_{Tk}$. For the experiment in the presence of L-Asp (brown) each point was calculated averaging the data from three technical replicates and then normalised dividing each averaged current by the maximum amplitude (I/I$_{max}$). For the experiment in the absence of L-Asp (red), each point was calculated averaging the data from three technical replicates

and normalised dividing each averaged current by the maximum amplitude measured in the experiment in the presence of L-Asp. Empty liposomes in the presence of Na$^+$ and L-Asp (black) were used as negative control. The error bars are representative of the standard deviation calculated on three biological replicates; **c** Peak currents at increasing concentrations of NaSCN or KSCN on the wild type in the presence or absence of L-Asp. The brown curve represents the fit on the peak currents measured in the presence of L-Asp, Na$^+$ and SCN$^-$ and averaged and normalised as explained in **b** for the condition in the presence of L-Asp. The red and cyan curves represent fit on the peak currents measured in the absence of L-Asp and the presence of NaSCN and the presence of L-Asp and KSCN, respectively and normalised as explained in (**b**) for the condition in the absence of L-Asp. The error bars are representative of the standard deviation calculated on three biological replicates. Source data are provided as a Source data file.

For this series of experiments, a Na$^+$ gradient was established by exchanging a resting solution containing K$^+$ with a non-activating solution consisting of increasing concentrations of Na$^+$ (R → NA). The resulting Na$^+$ gradient across the membrane is a prerequisite for L-Asp transport and anion conductance, but without amino acid substrate present, the experimental condition remains non-activating. Subsequently, the non-activating solution was replaced by the activating solution containing the L-Asp (NA → A). The proteoliposomes were subjected to a change from the resting solution (300 mM KPi buffer pH 7) to the non-activating solution (X mM NaPi pH 7 + (300-X) mM KPi pH 7) to the activating solution (X mM NaPi pH 7 + (300-X) mM KPi pH 7 + 100 μM L-Asp).

To validate the correct functionality of wild-type Glt$_{Tk}$ in this experimental setup, we measured the Na$^+$ dependence of L-Asp transport (Fig. 2a). Upon the NA → A exchange, the jump in the concentration of L-Asp led to a positive transient current in the presence of a Na$^+$ gradient. The positive current originates from the net transfer of two positive charges into the proteoliposomes resulting from the 3:1 Na$^+$:L-Asp coupling stoichiometry. The positive current jump amplitudes were dependent on the Na$^+$ concentration. They increased until ~50 mM Na$^+$ and levelled off at higher concentrations. By plotting the peak current amplitudes against the Na$^+$ concentration and using the Hill equation for fitting, we determined a K$_M^{app}$ for Na$^+$ of 12.3 ± 8.9 mM ($n$ = 1.5 ± 0.4), which is comparable to the apparent affinity measured at the same L-Asp concentration

(100 μM) with radiolabelled L-Asp uptakes[16] (Fig. 2a, top and Supplementary Fig. 2a).

The same setup was used to measure the Na$^+$ dependence of current resulting from the anion conductance. The proteoliposomes were subjected to a change from the resting solution (113.5 mM KPi pH 7 + 100 μM L-Asp) to the non-activating solution (X mM NaPi pH 7 + (113.5-X) mM KPi pH 7 + 100 μM L-Asp) to the activating solution (X mM NaPi pH 7 + (113.5-X) mM KPi pH 7 + 100 μM L-Asp + 10 mM SCN$^-$). SCN$^-$ was chosen as the anion in these experiments because of its higher permeability through Glt$_{Tk}$ in comparison with other anions (similar to what has been observed in EAAT1, EAAT2, EAAT3[47]) (Supplementary Fig. 4a). Both L-Asp and Na$^+$ were added in the non-activating solution because the anion currents created by SCN$^-$ are expected to be dependent on the presence of both the amino acid substrate and the co-transported cation. Increasing Na$^+$ concentration, generated by the solutions exchange in the non-activating and activating solutions, resulted in an increase of the amplitude of the measured current in the presence of protein (Fig. 2a, bottom). In contrast with the measurements in the absence of permeant anions, which led to positive currents, the currents were negative in the presence of SCN$^-$. Similar to the measurements on L-Asp transport (Fig. 2a, top), the amplitude of anionic currents depended on the Na$^+$ concentration with a K$_M^{app}$ for Na$^+$ of 30.7 ± 11 mM ($n$ = 0.7 ± 0.3) (Fig. 2a bottom, 2b and Supplementary Fig. 2b). The peak currents generated by SCN$^-$ in the absence of L-Asp, in the presence of L-Asp but without protein (empty

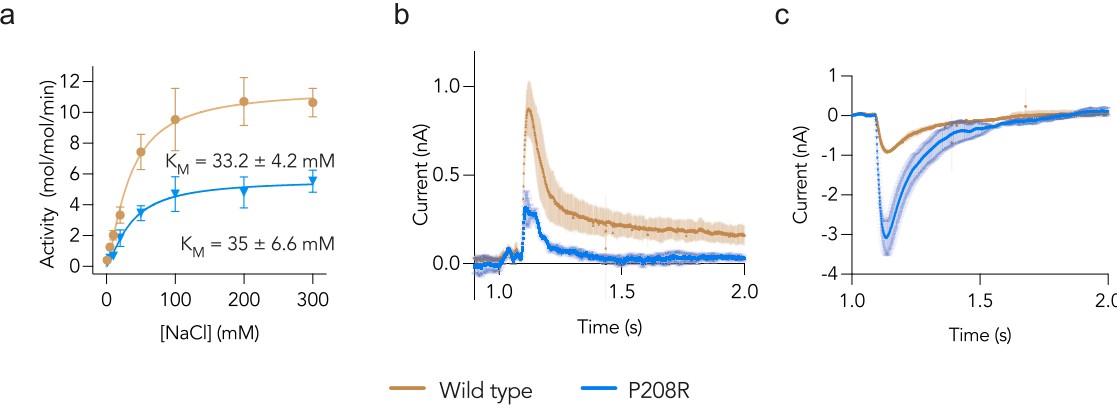

── Wild type ── P208R

**Fig. 3 | Impact of the P208R mutation on L-Asp transport and anion conductance. a** Sodium dependence of transport at the fixed concentration of 1 μM L-Asp in proteoliposomes containing wild type $Glt_{Tk}$ (brown) and the P208R mutant (blue), measured with using radioactive aspartate. The plots were fitted using the Hill model yielding a Hill coefficient of 1.36 ± 0.1 (WT) and 1.37 ± 0.2 (P208R). The error bars are representative of the standard deviation calculated on two biological and three technical replicates; **b** Comparisons of transient currents induced by 100 μM L-Asp at 200 mM Na⁺ on wild type and P208R. The bold lines are the average from two biological and two technical replicates. Error margins are indicated by the shaded areas and represent the standard deviation calculated on three biological replicates; **c** Comparisons of transient currents induced by 100 μM L-Asp and 10 mM SCN⁻ at 200 mM Na⁺ on wild type and P208R. The bold lines are the average from two biological and two technical replicates. Error margins are indicated by the shaded areas and represent the standard deviation calculated on three biological replicates. Source data are provided as a Source data file.

liposomes), or in the absence of Na⁺ were of low amplitude indicating that both L-Asp and Na⁺ are required for opening of the channel (Fig. 2b, Supplementary Fig. 4c). This result, together with similar Na⁺ dependence ($K_M^{app}$) of the L-Asp and SCN⁻-induced peak currents, is consistent with the notion that the anion channel requires the transition between outward facing state and inward facing states associated with the transport of L-Asp[20]. Furthermore, a double solution exchange (DSE) experiment using increasing concentrations of SCN⁻ showed that the absence of L-Asp or the use of KSCN instead of NaSCN (therefore in the absence of Na⁺) reduced the currents severely (Fig. 2c and Supplementary Fig. 4b). Altogether, these results indicate that similarly to the mammalian homologues and $Glt_{Ph}$ the opening of the anionic channel in $Glt_{Tk}$ is substrate-gated[4,21,32–34].

### P208R mutation affects substrate uptake and anionic current in $Glt_{Tk}$

To assess whether the P208R mutation affects the function of $Glt_{Tk}$, wild-type and mutant proteins were purified, reconstituted into proteoliposomes and the uptake of radiolabelled aspartate at different concentrations of Na⁺ was measured (Supplementary Figs. 5–7). Fitting the Hill equation to the data of both the wild type and the mutant afforded similar $K_M^{app}$ values for Na⁺ for the mutant and wild type protein (33.2 ± 4.2 mM and 35 ± 6.6 mM, respectively), while $V_{max}^{app}$ of ¹⁴[C] L-aspartate transport by $Glt_{Tk}$_P208R was reduced two-fold when compared to the wild type (11.4 ± 0.5 mol/mol x min⁻¹ vs 5.6 ± 0.4 mol/mol x min⁻¹ (Fig. 3a)). A similar reduction in transport rates was previously observed for the mammalian homologues.

We then used SSM to compare the currents generated by L-Asp transport and SCN⁻ on the wild type and the mutant proteins (Fig. 3b, c). In the L-Asp transport experiment measured by SSM, the amplitude of the current recorded for P208R at 200 mM Na⁺ was reduced by 2.5-fold compared to the wild type (Fig. 3b). Consistent with the [¹⁴C]-L-aspartate transport experiments, $K_M^{app}$ values for Na⁺ were not significantly different for the mutant and wild-type proteins (Supplementary Fig. 2a).

The SSM experiment for measuring SCN⁻ conductance on the two proteins revealed that, in contrast to the reduction of the L-Asp transport rate caused by the mutation (Fig. 3b and Supplementary Fig. 7b, top), the currents generated by SCN⁻ on $Glt_{Tk}$_P208R were increased by around 3-fold compared to the wild type (Fig. 3c, and Supplementary Fig. 7b, bottom), while $K_M^{app}$ for Na⁺ was not significantly affected by the mutation (Supplementary Fig. 2b).

### Conservation of architecture and binding site in the mutant

We then solved a structure of $Glt_{Tk}$_P208R embedded in nanodiscs and in the presence of Na⁺ and L-Asp using single-particle cryo-EM at 3.25 Å resolution (Table 1 and Supplementary Figs. 8–9). The structure revealed that all three protomers were in the fully outward

**Table 1 | Cryo-EM data collection, refinement and validation statistics**

| Data collection and processing | |
| --- | --- |
| Magnification | 130k |
| Voltage (kV) | 200 |
| Electron exposure (e⁻/Å²) | 51 |
| Defocus range | −0.5 to −2.0 |
| Pixel size (Å) | 1.022 |
| Symmetry imposed | C3 |
| Initial particle images (no.) | 1,211,030 |
| Final particle images (no.) | 701,140 |
| Map resolution (Å) FSC = 0.143 | 3.25 |
| Refinement | |
| Initial model | PDB 5E9S |
| Model composition | |
| Nonhydrogen atoms | 9166 |
| Protein residues | 1208 |
| R.m.s deviations | |
| Bond lengths (Å) | 0.008 |
| Bond angles (°) | 0.61 |
| Validation | |
| MolProbity score | 1.31 |
| Clashscore | 5.75 |
| Rotamer outliers (%) | 0.3 |
| Ramachandran plot | |
| Favoured (%) | 98.32 |
| Allowed (%) | 1.68 |
| Disallowed (%) | 0.0 |
| Rama-Z Whole/helix/loop | 0.4/0.39/0.36 |
| CC_mask | 0.84 |
| CC_volume | 0.79 |

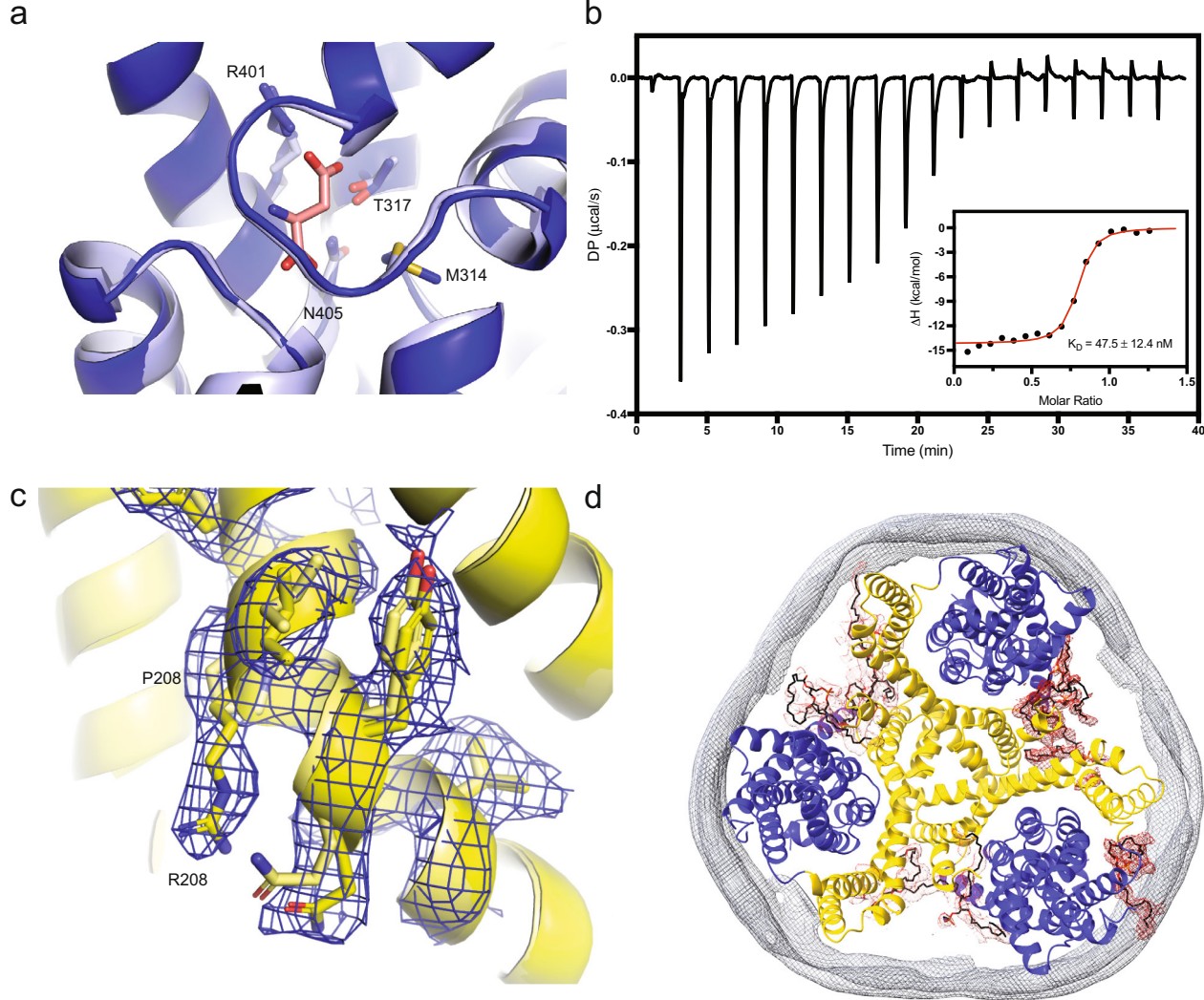

**Fig. 4 | Structural integrity of the P208R mutant.** The P208R mutation has no consequences to L-Asp binding as seen from (**a**) the comparison of binding sites between the mutant and WT (note that not only residues of the binding site but also the position of HP2 are similar) and (**b**) binding affinity measurements by ITC (the error is given by the standard deviation calculated on three independent experiments). Cryo-EM density (**c**) for the side chain of arginine 208 and neighbouring residues (view from the membrane plane, extracellular side below) and (**d**) for the belt protein of a nanodisc (grey) and putative lipids (red). Maps are shown at $4\sigma$. Source data are provided as a Source data file.

oriented state, rather than the intermediate outward state observed previously for the wild-type protein in the same conditions, indicating that the states may be very similar energetically and minor differences in the experimental procedure can favour different conformation of the transport domain. The fully outward position of transport domains has commonly been observed in structures of substrate-loaded glutamate transporters obtained with X-ray crystallography[12,48–50]. Therefore, the crystallographic structure of the wild type protein with the Na+ and Aspartate bound (PDB: 5e9s) was selected for comparison. At the obtained resolution, any change in the geometry of the L-aspartate binding site or the rearrangement of TM5 should be detectable[36]. The side chains of the residues in L-Asp binding site (Ser280, Met314, Thr317, Thr355, Arg401, Thr402, Asn405) assume the same position in the wild-type and mutant proteins (Fig. 4a), confirming that there is no rearrangement that leads to a disruption of the binding site. The conservation of the binding site is furthermore confirmed by ITC using the detergent-purified protein (Fig. 4b). The affinity of the mutated protein for L-aspartate was similar to that of the wild type at the same concentration of Na+ ($47.5 \pm 12.4$ nM)[16]. As shown in Fig. 4c, the mutant protein still has the kink at position 208 in helix 5,

despite removal of the helix breaking proline residue. Apparently, the architecture of the $Glt_{Tk}$ scaffold domain is more robust than previously thought[36]. The density of the arginine side chain points towards the lipidic environment (Fig. 4c and Supplementary Fig. 10). Although the resolution of the structure of the P208R mutant is not high enough to unambiguously identify phospholipids, it is likely that the arginine side chain interacts with lipid molecules (Supplementary Fig. 10). To test this hypothesis, we performed all-atom MD simulations of the P208R mutant embedded in a lipid bilayer composed of POPE, POPG and POPE (see Methods for details). Within the first hundred nanoseconds of simulation stable salt bridges are formed between the arginine side chain of each of the three protomers and the headgroups of negatively charged lipids (POPG). These salt bridges remained stable throughout the entire MD trajectory (of 900 ns). (Supplementary Figs. 11–12). Importantly the formation of this salt bridge is also compatible with the anion-conductive state, which we simulated via threading of $Glt_{Tk}$ P208R sequence onto the $Glt_{Ph}$ model in chloride conducting state (PDB ID: 6WKY) (Supplementary Fig. 13) and apparently leads to the pore widening (Supplementary Fig. 14 and Supplementary Movie 1).

## Discussion

In 2005, Jen and collaborators reported the case of a young boy affected by episodic ataxia 6 (EA6), a type of ataxia characterised by seizures, migraine and hemiplegia[37]. Through exons amplification and screening, they identified a non-inherited mutation in the gene encoding the glial glutamate transporter SLC1A3 (hEAAT1) as the cause of EA6. While wild-type hEAAT1 primarily acts as secondary active transporter that translocates glutamate or aspartate across the membrane, with relatively minor amino acid-gated anion channel activity[2,5,7–9,17–22], fluorescence lifetime imaging microscopy data of Bergman glial cells expressing the mutant revealed that these cells had an increased chloride efflux that causes a reduction in cell volume and apoptosis[42]. Also, in homologues of hEAAT, the corresponding mutation is responsible for a lower transport of glutamate and a higher anion conductance also in other homologues[36,37,41]. From a molecular point of view, in the disease-linked version of the protein, the only proline present in the TM5, proline 290, is mutated into an arginine. Together with other residues located between the scaffold and the transport domain, this proline forms the chloride permeation pathway[33,34,51,52]. This aqueous pore becomes accessible during the elevator movement with a mechanism that has been described either as lateral movement of the trimerization domain[21], or as gated by two hydrophobic gates[22].

Prolines have no hydrogens on main chain nitrogen; hence they cannot act as hydrogen bond donors in α-helices and β-strands. Therefore, prolines are mostly found either in loops[53] or at the edges of β- sheets and the start of α-helices, where there is no requirement for hydrogen bond donors in the backbone. However, if placed into a secondary structure element, prolines normally lead to formation of a kink. Proline 290 in EAAT1 (and in the corresponding positions in other glutamate transporter homologues[36]) is indeed located in the characteristic kink of transmembrane helix 5 located in the trimerization domain (Fig. 1b). It has been postulated in the literature that the mutation to arginine at this position most likely leads to the removal of the kink in helix 5 and therefore promotes a change of the overall protein architecture, affecting the binding sites for glutamate/aspartate and Na+ [36,37].

In the present work, we investigated the functional and structural effects of the point mutation from proline to arginine at position 208 on the archaeal homologue Glt$_{Tk}$. In order to validate it as a model for this study, we confirmed the existence of an anion channel in this protein using solid-supported membrane electrophysiology. While L-Asp transport leads to a Na+-dependent positive current, similar to the one measured with other experimental setups[16], the conductance of SCN- creates a negative current which is strictly dependent on L-Asp, and has the same sodium dependence (Fig. 2). Therefore, in addition to EAAT1, EAAT2 and Glt$_{Ph}$, this work has confirmed that the opening of the channel of Glt$_{Tk}$ is indeed substrate-gated. This corroborates once again the high degree of conservation of the chloride channel and its molecular determinants[21,32].

The radiolabelled L-Asp uptakes as well as double-solution exchange experiments on SSM showed that the mutant Glt$_{Tk}$_P208R transports the substrate L-Asp with a reduced transport rate compared to the wild type (Fig. 3a, b). In sharp contrast to the reduced transport of L-Asp, the mutant P208R shows increased SCN− current amplitudes compared the wild type (Fig. 3b, c). Therefore, most likely this mutation favours an anion conducting conformation leading to reduced L-Asp transport. Moreover, the affinity of the mutant for L-Asp is within the same range of the wild type, indicating that there is no change in the geometry of the binding site (Fig. 4a, b). This result is confirmed by the structure of Glt$_{Tk}$ embedded in nanodiscs that were obtained at 3.25 Å resolution using single particle cryo-electron microscopy (Supplementary Fig. 9). Within this structure, no major rearrangements of the protein were observed, and the binding site exhibits the same geometry as the wild type. Also, despite the replacement of

proline to a positively charged arginine, the kink in helix 5 is maintained. Thus, altered activity of the mutant protein is not related to observable changes in the protein architecture, and must be caused by more subtle effects.

The unambiguous density of the arginine side chain (Fig. 4c) pointing towards the lipidic environment suggests an interaction between the polar side chain and the heads of the phospholipids (Supplementary Fig. 10). MD simulations confirmed that such a salt bridge remains long-lived (Supplementary Figs. 11–12). Perhaps the interaction with the lipid leads to longer dwell times in the anion-conductive state, and somewhat disfavour the transition to the inward state, which is then reflected in the reduced transport rate and increased anion conductance. The observations that this mutation neither disturbs the binding site geometry, nor affects the binding affinities, yet diminishes the L-Asp transport with simultaneous increase in anion-conductance, support this explanation. Another possibility is that the salt bridge between the arginine and the phospholipid might widen the anion channel and facilitate the passage of anions during the turnover, and this is reflected in the increase of calculated solvent density map for the P208R mutant (Supplementary Fig. 14, Supplementary Movie 1). This would imply the lateral movements of domains (or at least a dislocation of the transport domain) take place during the transition between the outward facing state (OFS) and the inward facing state (IFS), when the anion channel is expected to occur[54]. Such rearrangements are not observed in the structure reported here as it is in OFS, but can be inferred from our MD simulations. Further investigations on hEAAT1 in anion conducting state are required to unambiguously answer this question.

In conclusion, we have shown that the mutation of a helix-breaking proline into arginine does not abolish the kink, invalidating previous hypotheses that such a mutant would undergo major structural rearrangements affecting the binding site and the transient anion channel. Given the high degree of structural and functional conservation in this family of transporters, we can conclude that such a mutation has no impact on the global organisation of transmembrane helices still diminishing the transport and favouring the anion-conductive state via protein-lipid interactions.

## Methods

### Cloning

In order to mutate proline 208 into arginine, a site directed mutagenesis using a PCR-cloning strategy was used. The two primers 5′-TTATGCAGTATGCACGGATTGGTGTTTTTG- 3′ (forward) and 5′- AAAAACACCAATCCGTGCATACTGCATAA − 3′ (reverse) were used to amplify and modify the gene of Glt$_{Tk}$ in the plasmid pBAD24. One unit of DpnI (20 U/μL, New England Biolabs) was added to the reaction to remove the methylated DNA. Following this, *E. coli* MC1061 chemocompetent cells were transformed with the mutated plasmid using heat shock. The cells were plated on LB-agar medium containing 100 μg/mL ampicillin for selection. Several colonies were picked and grown overnight at 37 °C while shaking at 220 rpm. The day after, cells were harvested and plasmid DNA was purified using a NucleoSpin miniprep kit. The presence and the correct position of the mutation were verified through the Sanger method by EurofinsGenomics service.

### Protein expression

A single colony of *E. coli* MC1061 containing pBAD24_glttk_P208R was grown in LB medium with 100 μg/mL ampicillin at 37 °C while shaking at 220 rpm. When cells reached the OD$_{600}$ of 0.9 they were supplemented with 0.05% of L-arabinose. After 3 h of induced expression, cells were harvested by centrifugation for 15 min at 7400 × g. All the steps that followed were performed at 4 °C. The pellet was then resuspended in 20 mM Tris-HCl, pH 8.0, 1 mM MgSO$_4$, 100 μM Dnase I and 200 μM PMSF. The suspension was passed through a high-

pressure homogeniser (HPL6, Maximator) at 25 kPsi. Membrane vesicles were then isolated by ultracentrifugation (90 min, 193,360 × $g$) and homogenised using a potter in a buffer containing 20 mM Tris-HCl, pH 8.0.

## Purification of *apo* Glt$_{Tk}$ and Glt$_{Tk}$_P208R

For purification of *apo* proteins, one aliquot of membrane vesicles was solubilised in 50 mM Tris-HCl pH 8.0, 300 mM KCl, 1% (v/v) n-dodecyl-β-D-maltoside (DDM) for 45 min at 4 °C. The insoluble fraction was removed by ultracentrifugation for 30 min at 265,000 × $g$ at 4 °C and the soluble fraction was supplemented with 15 mM imidazole pH 8.0 and incubated with 0.5 mL Ni$^{2+}$-sepharose resin (GE Healthcare) on a rocking platform at 4 °C for 1 h. Following this, the unbound fraction was allowed to flow through a Poly-prep column (Biorad) and the column was washed with 50 mM Tris-HCl pH 8.0, 300 mM KCl, 60 mM imidazole pH 8.0 and 0.15% (v/v) n-decyl-β-maltoside (DM). Then, the protein was eluted in the same buffer containing 500 mM imidazole pH 8.0. The eluted protein was further purified from imidazole by size exclusion chromatography on the Superdex 200 10/300 column (GE Healthcare) in the gel filtration buffer 10 mM HEPES-KOH, pH 8.0, 100 mM KCl and 0.15% DM.

## Reconstitution into proteoliposomes

An aliquot of 20 mg/mL *E. coli* polar lipid extract in 50 mM potassium phosphate buffer (KPi) pH 7.0 was extruded by 11 passages through a 400 nm diameter polycarbonate filter (Avestin). After diluting the lipid mixture to 4 mg/mL using the same buffer, it was destabilised with 10% Triton-X100 following the absorption at 540 nm (Jasco). The titration was stopped once the absorption signal decreased to about 60% of the maximum value reached. The two protein variants were then separately added to the solutions in two different protein:lipid ratios: 1:800 (w/w) for the radioactive assays and 1:75 (w/w) for SSM, and incubated for 30 min at 20 °C on a rocking platform. In order to remove detergent, Bio-beads were increasingly added to the mixture and incubated at 4 °C in this order: 25 mg/mL for 30 min, 15 mg/mL for 1 h, 19 mg/mL overnight and 4 mg/mL for 2 h. After removing Bio-beads, the proteoliposomes were centrifuged at 298,906 × $g$ at 4 °C for 20 min, resuspended to 20 mg/mL in internal lumen buffer (different for radioactivity measurements and SSM – see the following paragraphs). Internal buffer was exchanged by three cycles of freeze-thawing. Finally, the aliquots were stored in liquid nitrogen until further use.

## Cryo-EM sample preparation and data acquisition

*E. coli* lipids (the same used for proteoliposomes) were destabilised with 30 mM β-DDM, left shaking at 20 °C for 30 s and incubated on a rocking platform for at least 3 h. The purified Glt$_{Tk}$ variants were mixed with the destabilised lipids in a 3:5 molar ratio (protein:lipids) and left on ice for 30 min. Then the belt protein MSP2N2 (purified beforehand in 20 mM Tris-HCl, 100 mM KCl, pH 8 and cleaved from the His-tag) was added to the mixture in order to have a final molar ratio of 3:5:100 (Glt$_{Tk}$:MSP2N2:lipids). The mixture was left to incubate on a shaker at 4 °C for 90 min. After that, to remove detergent, 700 mg/ml BioBeads were added to the mixture and left incubate overnight. Nanodiscs were formed upon removal of the detergent. The day after the BioBeads were removed, the mixture was left to bind 0.5 mL Ni$^{2+}$ sepharose resin pre-equilibrated with 20 mM Tris-HCl, 100 mM KCl, pH 8 for 1 h. The nanodiscs and resin were then pooled through a column and washed with the equilibrating buffer containing 30 mM imidazole. The nanodiscs were purified in 500 mM imidazole. The eluted nanodiscs were then further purified from imidazole by size exclusion chromatography on the Superdex 200 10/300 column (GE Healthcare) in the gel filtration buffer 20 mM Tris-HCl, pH 8.0, 100 mM KCl (Supplementary Fig. 5). Prior to the preparation of the grids the nanodiscs were concentrated up to 1.5 mg/ml and 300 mM NaCl and 100 µM L-Asp were added to the mixture. Both grids preparation (using Quantifoil grids

(Au R1.2/1.3, 300 mesh)) and data acquisition were performed at NeCEN facility using Vitrobot-IV (Thermofischer) and Titan Krios, equipped with K3 detector, respectively. The data processing and model building/refinement were performed in Cryosparc and Coot/phenix.real_realspace.refine, respectively. Image processing and model building workflows are summarised in Supplementary Fig. 8. The refinement statistics and model quality are reported in Table 1. To model the ClCS we used One to One threading mode of Phyre2 server[55] using Glt$_{Ph}$ model as a template (PDB ID 6WYK).

## Uptake assay

Proteoliposomes were thawed and collected by centrifugation for 20 min at 298,906 × $g$ at 4 °C. The pellet was resuspended in the internal lumen buffer containing 10 mM KPi, pH 7.0, 300 mM KCl and extruded by 11 passages through a polycarbonate filter (400 nm, Avestin). The extruded proteoliposomes were centrifuged again using the same conditions as in the previous step and resuspended in the same buffer to a concentration of 100 mg/mL. To record initial rates of transport at different external Na$^+$ concentrations, 2 µL of proteoliposomes were diluted 100 times in the reaction buffers containing: 10 mM KPi, pH 7.0, X mM NaCl, (300 mM – X) mM Choline Chloride, 500 nM valinomycin, 500 nM L-aspartate, 500 nM $^{14}$[C]-L-aspartate. After 30 s from the start of the experiment, the 200 µL reaction mix was quenched in 2 mL of ice-cold buffer (10 mM KPi, pH 7.0, 300 mM KCl) and filtered through a nitrocellulose filter (Protran BA 85- Whatman filter). These steps were repeated for all the different concentrations of NaCl (5, 20, 50, 100, 200 and 300 mM) and filters were dissolved in a 2 mL scintillation cocktail (Ultima Gold MV from PerkinElmer). Radioactivity was then measured by counting each point for 2 min using a PerkinElmer Tri-Carb 2800RT scintillation counter. See Supplementary Fig. 6 for the estimation of the concentrations of reconstituted proteins (WT and P208R) in proteoliposomes for L-Asp transport uptakes.

For the time-course experiment to determine the transport efficiency in the proteoliposomes at different lipid:protein ratios (Supplementary Fig. 7a), the procedure was the following: for every point, 2 µL of proteoliposomes were diluted 100 times into 198 µl of reaction buffer (10 mM KPi, pH 7.0, 300 mM NaCl, 500 nM valinomycin, 500 nM L-Aspartate, 500 nM [$^{14}$C]-L-Aspartate). At the start of the experiment (time point 0), the 200 µl reaction was quenched in 2 mL ice-cold buffer (10 mM KPi, pH 7.0, 300 mM KCl) and filtered through a nitrocellulose filter. These steps were repeated for all the other time points (1,2,3,4 min) and filters were dissolved in the scintillation cocktail and radioactivity was measured as explained above.

## Solid-supported membrane experiments

Liposomes or proteoliposomes were absorbed onto an artificial bilayer pre-formed on a gold surface and transport currents were detected via capacitive coupling, as described elsewhere[56]. Briefly, gold sensors were incubated with 50 µL of 1 mM octadecanethiol solution and kept at 20 °C for at least 30 min; after incubation, the thiol solution is removed through a wash with milli-Q water and 100% isopropanol. During this step, a hydrophobic monolayer is formed on top of the gold surface. To create the bilayer, a 1 µL droplet of lipid solution (diphytanoyl phosphatidylcholine: octadecylamine 60:1 in n-decane) was added to the surface, followed by 50 µL of resting solution containing either 300 mM KPi or 113.5 mM KPi, pH 7 buffer and then left to incubate for at least 30 min at room temperature. A fraction of proteoliposomes was thawed, extruded 11 times through a 200 nm polycarbonate filter, centrifuged at 298,906 × $g$ at 4 °C and resuspended to a final concentration of 5 mg/mL. 10 µL of the suspension was added to the sensors that were then centrifuged at 2500 × $g$ for 30 min, in order to ensure the complete adhesion of the proteoliposomes to the surface. The sensor was then placed in a chamber where the solution exchange is performed by a software-

controlled experiment programmed in SURFE²R (Nanion). All the buffer exchanges are listed in the figures and their legends. Each perfusion in the non-activating and in the activating buffers lasted one second and the flow rate of the measurement was 200 μL/s. In line with the guidelines, only sensors with a capacitance in the range of 15–30 nF and conductance below 5 nS were considered for the measurements[45]. Each double-solution exchange measurement was repeated on each sensor at least three times. The average of the three measurements was then considered for the comparison with other sensors. See Supplementary Fig. 6 for the estimation of the concentrations of reconstituted proteins (WT and P208R) in proteoliposomes for SSM measurements.

## ITC

Binding affinity of the detergent purified mutant $Glt_{Tk}$_P208R for L-Asp was measured using isothermal titration calorimetry (MicroCal PEAQ-ITC, Malvern Panalytical). In order to allow purified protein to bind to the substrate, 300 mM NaCl was added to purified protein (in gel filtration buffer, 10 mM HEPES-KOH, pH 8.0, 100 mM KCl and 0.15% DM) and the protein concentration was adjusted to 15 μM. The gel filtration buffer supplemented with 300 mM NaCl was also used to adjust the concentration of the substrate L-Asp to 100 μM. The protein was then added to the pre-equilibrated cell and kept at 25 °C while stirring at 750 rpm. The substrate was incrementally titrated into the cell in 20 injections of 2 μL each at intervals of 120 s. Data were fitted using the MicroCal PEAQ-ITC Analysis Software.

## Molecular dynamics simulations

The starting protein model was taken from a structure of glutamate transporter homologue $Glt_{Tk}$ in unsaturated conditions - inward-inward-outward configuration (3.39 Å resolution, PDB ID: 6XWO[9] for the wild-type and the structure reported here for the mutant (P208R) as well as the model obtained from Phyre2 server for anion conducting state. Atomistic MD simulations of $Glt_{Tk}$ WT and mutant (P208R) embedded in a lipid membrane in aqueous salt solutions were performed. All MD simulations presented in our work have been completed using the MD package Gromacs (version 2018.1)[57] and the CHARMM36 force field[58].

The membrane builder tool of the CHARMM-GUI[59] was used to embed the protein structure in a rectangular lipid bilayer composed of a mixture of 397 1-palmitoyl-2-oleoyl-sn-glycero-3-phosphoethanolamine (POPE), 1-palmitoyl-2-oleoyl-sn-glycero-3-phosphoglycerol (POPG) and 1-palmitoyl-2-oleoyl-sn-glycero-3-phosphocholine (POPC) lipids in the ratio 3:3:2 as was used in the liposomes in our in-vitro experiments and solvated with 150 mM aqueous NaCl solution. The box dimensions of the system are $130 \times 130 \times 120$ Å. The system was then solvated with TIP3P water molecules[59] such that every protein atom was at least 12 Å away from the side of the box. Periodic boundary conditions were employed and the particle-mesh Ewald method[60,61] was used for treatment of long-range electrostatic interactions. The systems were optimised and equilibrated for 1 ns in the NVT ensemble and 20 ns in the NPT ensemble. After the equilibration stages, 750 ns (WT) and 850 ns (P208R) long unrestrained runs were carried out. The pressure was maintained at 1 atm semi-isotropically with the Parinello–Rahman barostat[62] and a coupling constant of 1.0 ps. The simulations were conducted at a constant temperature of 303.15 K using the Nosé-Hoover thermostat[63,64]. The total number of atoms in the simulation box was ~207,000 atoms.

Visual inspection of the trajectories was performed with VMD[65] and PyMOL (Schrödinger).

## Reporting summary

Further information on research design is available in the Nature Portfolio Reporting Summary linked to this article.

## Data availability

The final model of P208R mutant and the cryo-EM map are deposited in PDB and EMDB databanks under 8AFA and EMD-15393 respectively. Source data are provided with this paper.

## Code availability

All models and parameter files for performed MD simulations are freely available from the Zenodo website (https://doi.org/10.5281/zenodo.6811055).

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

## Acknowledgements

The access to NeCEN facilities was funded by the Netherlands Electron Microscopy Infrastructure (NEMI), project number 184.034.014 of the National Roadmap for Large-Scale Research Infra- structure of the Dutch Research Council (NWO). We thank the personnel of NeCEN for the help with data collection. This research was financed by NWO grant OCENW.KLEIN.141 to A.G. and by European Union's Horizon 2020 research and innovation programme under the Marie Skłodowska-Curie grant agreement No: 860954.

## Author contributions

E.C. cloned, expressed and purified proteins; performed radioactive uptakes. E.C. and Z.R.A. performed ITC measurements. E.C., Z.R.A. and M.F.P.-R. performed SSM measurements. M.N. performed MD simulations. J.W. processed EM data, built and refined the model. E.C., D.J.S. and A.G. wrote the manuscript. All authors have given approval for the final version of the manuscript.

## Competing interests

The authors declare no competing interests.
