## [Peer Review File · Nature Communications]

Mutation in glutamate transporter homologue GltTk provides insights into pathologic mechanism of episodic ataxia 6Reviewers' comments:

Reviewer #1 (Remarks to the Author):

Summary

Excitatory amino acid transporters (EAATs) are a family of trimeric secondary active transporters with anion channel activities which play important roles in synaptic signal transmission. The archaeal homologs GltPh and GltTk are well-established structural and functional model systems for understanding EAAT physiology. In human EAAT1 (hEAAT1), the P290R mutation has been associated with episodic ataxia 6 and produces an increased anion conductance. In this work, the authors use the homologous P208R GltTk mutant to pursue a mechanistic basis for episodic ataxia 6. They impressively use complementary assays to measure anion conductance, cation coupling, and L-Asp binding in WT and P208R, and further solve a cryo-EM structure of the mutant. Functional assays indicate reduced L-Asp transport and an increase in the anion:L-asp transport ratio, as was seen in hEAAT1, but the structure surprisingly showed essentially no differences from a WT one. The authors speculate that Arg-lipid interactions stabilize the helical kink, but could allow increased anion conductance during turnover.

Validity

The use of complementary techniques is impressive, and helps bolster the findings from each individually. The agreement with the previous characterization of GltTk lends further confidence to the results. I do feel that overall validity is harmed somewhat by some missing data in the current form (see below), for both biochemical and structural aspects. These can mostly be resolved by changing the visualizations.

Significance

This manuscript describes an impressive set of biochemical and structural investigations which demonstrate the value of GltTk as a model for a serious human disorder, and could potentially lead to clinically useful insights into EA6. The failure to identify obvious structural correlates of modified function in the P208R structure, although quite interesting in itself, limits this somewhat, as the proposed mechanism is essentially speculative and relies on MD and changes in unobserved states.

Specific comments about data and analysis

SSM experiments

The SSM experiments are in general adequately described and appropriately conducted, but some details and important data are missing. Although the authors cite Krause et al 2008, where mEAAC1 is extensively characterized similarly with SSM electrophysiology, the two proteins do have different transport mechanisms (GltTk lacks the H⁺ and K⁺ coupling of mEAAC1), and so there are still some questions about how differently they will behave.

1 - Experimental details for SSM perfusion should be expanded. The flow rates, times for each perfusion step, QC ranges for capacitance and conductance, and details about how many times each sensor was used aren't clearly indicated.

2 - The normalization process in panel 2 B should be explained.

3 - Full traces for the double exchange experiments should be shown somewhere, even if just in the SI. The representative traces in panel 2 A appear to show the NA and A phases but not the perfusion with R buffer. Also, how many were performed?

4 - Full traces for the empty liposome experiments should be included. Also, it appears that portions of the +L-Asp trace in SI Figure 2 B and the 300 mM NaSCN trace in SI 2 C are cut out - these should be shown.

5 - What are the samples and experimental conditions in SI Figure 2 A?

6 - What concentration was the NaSCN jump in SI Figure 2 B? What concentration was L-Asp?

Data fitting

The KM values are fit to data while allowing the Hill coefficients to vary freely between mutant and WT. The logic behind this should be more explicit (should cooperativity vary between them?), or some sensitivity analysis should be performed (do KM values change if n is fixed?).

Data interpretation

The claims about the effects of P208R on anionic conductance are slightly confusing to me. Line 343 says the "currents generated by NaSCN on GltTk_P208R were not significantly different from the ones in the wild type." The Figure 3 legend says "P208R mutation...increases the anion current amplitude." The

relative amplitudes are not compared as far as I can see, as Figure 2 B compares normalized currents and finds the Na-dependence is indistinguishable. This should be clarified.

L348: I am not sure one can infer anything about relative open probabilities from these results. I am convinced that the relative anion to L-Asp conductance is increased in P208R, but the results as shown seem to imply diminished L-Asp transport and identical anion conductance. A different experiment would be necessary to make a claim about P(open).

Cryo-EM structure determination

I am not a cryo-EM expert, but I believe some necessary details are missing from the text and SI.

1 - The local resolution needs to be shown, not just the FSC map.

2 - The orientation preferences of particles should be shown.

3 - Does the second class during 3D classification post-ab initio represent junk particles or alternative states?

4 - Some details about grid type and freezing conditions would be nice, but not necessary.

Minor notes

Figures 2, 3 - The error bars for the SSM and radioactivity data are not defined. Are they SEM, SD, a confidence interval? How were the uncertainties calculated, as well?

Figure 4 - What level are the densities being contoured at?

SI Figure 2 D - the color scheme here is a bit confusing, with the gray tones not following the concentration gradient.

Structure refinement - Calculating the Rama Z-score would be useful, as 0 outliers could indicate overrefinement.

Suggestion for clarity

A cartoon representation of the experimental schemes for the SSM experiments could help the reader follow them in Figures 2 and 3.

A cartoon for the proposed scheme in Figure 4 would also help the reader integrate the MD results with the overall functional data.

References

L330: A citation for the claim about reduced transport rates in mammalian homologs would be appropriate.

Reviewer #2 (Remarks to the Author):

Review Guskov Manuscript

Members of the EAAT family of transporters play critical roles in regulating the concentration of the neurotransmitter glutamate in the mammalian central synapse. In addition, mutations in this transporter have been shown to cause a rare genetic disease called episodic ataxia type 6 (EA6) primarily affecting the cerebellum. Previous work has examined the functional effects of this mutation on mammalian EAAT family members, but the structural and mechanistic consequences of the mutation have yet to be understood.

Here, Colucci and co-authors set up a model system to study the EA6 mutation using a bacterial homolog of the EAATs, GltTk. They demonstrate that GltTk shows analogous functional changes due to the mutation to those in the EAATs, then determine the structure of the mutant protein and perform molecular dynamics simulations to probe the mechanism of the change.

This paper is essentially divided into two parts; in the first the authors nicely establish assays for the transport activity and anion conductance of GltTk using solid-supported membrane (SSM) measurements. The assays reported here are robust and the data strongly support the idea that the GltTk mutation P208R has parallel effects to the EA6-causing mutation in EAAT1. In the second part of the paper the authors solve the structure of GltTk carrying the P208R mutation and perform some molecular dynamics calculations. Unfortunately, since the structure is solved in an outward-facing conformation which is not the anion-conducting state, these results fail to yield insight into the mechanism of the changes caused by the mutation and do not support the claim in the abstract that the

salt bridge between the introduced arginine and lipid is “responsible for the widening of the anionic channel.”

In principle, the functional assays reported here would be of interest to a specialty journal audience, but the mechanistic claims are completely unsupported by evidence.

Major Concerns:

1. The structural and molecular dynamics data do not support the conclusion that the anion channel is widened to cause the increase in anion conductance.

Recent reports from the Ryan lab clearly demonstrate the EAAT anion channel is formed when the transporter is in an intermediate state in the transport process close to the inward-facing state and quite different from the outward-facing state solved here. Indeed, there is no evidence of an anion channel in the original outward-facing structures of GltPh. The authors here make no attempt to predict the equivalent state of the GltTk protein; indeed, their only comment about the anion pore in the structure is the statement “In principle these interactions could lead to a change in the channel geometry during turnover.” Thus, their claim in the abstract about “widening of the anionic channel” is without any experimental basis. Similarly, though the simulations seem to support the idea that the protein-lipid salt bridge is stable, they do not provide insight into the nature of the anion pore.

2. The authors argue that the anion conductance is increased in P208R; however, their SSM data contradict this claim. Although they show that sodium and L-aspartate are linked to anion conductance in the wild type, the mutation does not seem to impact it.

Minor:

1. Figure 1B is introduced on page 2 (84) for the first time without referring to residue P208. The reader is confused about the residue number since only P290 of EAAT1 is introduced.

2. In some figures, the structural representation should be improved to create a better visual aid for the reader. For example, figure 1B could be more prominent and less transparent. In figure 4C, please clarify the orientation shown. The zoom in figure 4A obstructs the structure.

3. In the SSM current traces in figure 2, there seems to be a clear time difference between the peak of L-aspartate vs. anion-induced currents. It would be nice if the authors discussed potential reasons for the shift.

4. Between figures 2 and 3, the color scheme should be consistent for mutant and wild type, specifically wild type data in Fig.2B, compared to the rest. Also, it is not clear why the anion conductance data in Figure 2C are inverted relative to the equivalent data in the bottom of Figure 3B. Both should either be normalized as in Figure 3 or directly shown as in Figure 2.

5. Line 173 the molar ratios of GltTK:lipid:MSP is reported as 3:5:100. I suspect this is erroneous and should be either 3:100:5 or 5:100:3, as I don't expect that there are 100 MSPs per nanodisc with 5 lipids.

6. In several places, there are typos that could be improved (e.g., page 1(17) , ... a reduction in the substrates substrate transport, ...)

7. The authors should be consistent with American vs. British spelling in text and figure (e.g., leveled vs. levelled or normalized vs. normalised)

Reviewer #3 (Remarks to the Author):

EAAT transporters, especially expressed in neural cells, have double function of Na⁺-L-Glu symport and Cl⁻ conductance. Pro to Arg mutation in the TM5 reduces Glu transporting activity while Cl⁻ efflux increased, which causes reduction in neural cell volume and apoptosis, associated with a disease called episodic ataxia 6 (EA6). This manuscript uses archaeal homologue of Asp transporter Glt(TK), belonging to the same SLC1A family to elucidate the functional mechanism of anion conductance and of EA6. First, the authors performed solid-state membrane electrophysiology to confirm that the archaeal homologue can be used as model of EAATs. Then, they confirmed mutation of Pro208 to Arg in Glt(TK) elevated SCN⁻ conductance, while Asp transporting activity was reduced. Next, the authors solved the Cryo-EM structure of Glt(TK)_P208R mutant at 3.25 Å overall resolution. The structure showed that the mutation remains the kink of TM5 as WT, but Arg208 provides tight electrostatic interaction with the phosphate head group of lipid. MD simulation revealed that the interaction causes lateral movement of TM5, leading to the widening of anion channel.

The functional and structural analyses are solid. However, this reviewer questions why the authors did not use human EAAT transporters for the functional and structural analyses, instead of the archaeal homologue, since the Cryo-EM structures of human EAATs have already been solved. This is a major concern by this reviewer.

In addition to this concerns, this reviewer raises the following concerns as follows.

1. In the abstract, the authors described archaeal homologue of glutamate transporters Glt(TK). This description is misleading, since Glt(TK) is an Asp transporter.

2. For widening of anion channel in MD simulation, did the authors not observe any structural change of the channel in Cryo-EM structure ? Density of anion such as SCN⁻ could not be observed ? At least more detailed description for anion channel widening by MD simulation should be described.

3. Comparison of the P2087R mutant with that of WT Glt(TK) did not reveal structural change such as lateral movement of TM5 ?

Reviewer #4 (Remarks to the Author):

The manuscript by Colucci et al. uses a homolog of human glutamate transporters, the archaeal Glt^{Tk} to functionally and structurally characterize a proline to arginine mutation that is related to the episodic ataxia type 6 (EA6) neurological diseases. First, they use solid-supported membrane electrophysiology, Isothermal Titration Calorimetry (ITC), and radioactively labeled substrate uptake assays to prove that the Glt^{Tk} proline to arginine (P208R) mutant is a good model of the human glutamate transporter mutated in EA6 disease. Then, they used single-particle cryoEM to solve the structure of the Glt^{Tk} mutant. This structure reveals a similar kink of the helix at the position of the point mutation and the arginine side chain protruding toward the membrane bilayer, suggesting its interaction with the lipids headgroups. To prove this interaction, the authors performed a molecular dynamics simulation of the mutant and observed a persistent interaction between the arginine and the lipid headgroups.

The information provided in the manuscript could be of interest to the field of the glutamate transporter, but some issues need to be addressed to establish the validity of the presented conclusions. I will focus mainly on the computational results, which is my area of expertise while noting some issues from the experimental section.

1)- It is concluded that the ITC shows similar substrate binding affinities of the wild type and mutant, suggesting that the binding site does not change upon mutation. However, even if both dissociation constants are in the nanomolar range, the mutant K_D measured in this work is ~11 nM while the wild type, measured also by ITC in a previous article, is 75 nM (at the same concentration of sodium), almost 7 fold difference. The measurements were performed in slightly different conditions, i.e. different buffers, and different instruments, thus it might be worth measuring again the wild-type K_D to compare with the mutant values presented in this work.

2)- Why did they perform the simulations using the wild type inward-inward-outward trimer? But also, it might not be the fairest comparison with their fully outward structure. Maybe simulations of the outward-occluded crystal structure or the TBOA fully outward cryo-EM structure are better controls for the mutant simulation.

3)- From the RMSD progression, some protomers do not seem very stable in their simulation. Especially the one shown in the bottom pane of Fig S7 for the wild-type simulation. Does that correspond to the outward-facing protomer? Can you please discuss which kind of movements corresponds to the larger RMSD values?

4)- In the simulations, the authors used the same lipid mixture (3 POPE: 3 POPG: 2 POPC) present in the nanodisc. However, lipid mixtures converge slowly (in the order of hundreds of nanoseconds). Hence, they should indicate how they placed lipids in the bilayer (i.e. are their positions randomized? How?). They also need to indicate whether the lipids are equilibrated or not.

5)- The simulations might need to be repeated from different starting points to establish the robustness of the observed interactions (maybe a total of three simulations per system).

6)- Since the arginine-POPG interaction is not reversible in their simulation time scale, the authors should indicate whether the arginine in the P208R mutant binds to one specific POPG molecule or it exchanges freely with different POPG lipids in the simulation box.

7)- The authors speculate that this interaction between the arginine and the lipid headgroup might contribute to the widening of the chloride channel during the elevator movement. Do they observe more waters around the suggested pore in the simulations of the mutant, compared to the same conformational state of the wild-type?

Minor issues:

- Can you please discuss or speculate possible reasons why the mutant seems to stabilize the fully outward symmetric trimer? This conformation is not observed for the wild-type Glt^{Tk} under a similar experimental setup, even in saturated aspartate conditions (Arkhipova *et al.*, 2020 Nat Comm), so it might be a result of the P208R mutation.

- Can you compare the newly reported structure with the TBOA fully outward protomers?

- In the introduction section they mentioned that the ITC and the substrate uptake measurements showed reduced transport of the substrate, they should list only the uptake assay; the ITC showed only equilibrium binding.

- 6XWO is a cryoEM structure, not a crystal structure, please correct that in the methods section.

We thank all reviewers for their constructive criticism.
Below we give point to point response to the reviewers' comments

Reviewer #1 (Remarks to the Author)

Summary

Excitatory amino acid transporters (EAATs) are a family of trimeric secondary active transporters with anion channel activities which play important roles in synaptic signal transmission. The archaeal homologs GltPh and GltTk are well-established structural and functional model systems for understanding EAAT physiology. In human EAAT1 (hEAAT1), the P290R mutation has been associated with episodic ataxia 6 and produces an increased anion conductance. In this work, the authors use the homologous P208R GltTk mutant to pursue a mechanistic basis for episodic ataxia 6. They impressively use complementary assays to measure anion conductance, cation coupling, and L-Asp binding in WT and P208R, and further solve a cryo-EM structure of the mutant. Functional assays indicate reduced L-Asp transport and an increase in the anion:L-asp transport ratio, as was seen in hEAAT1, but the structure surprisingly showed essentially no differences from a WT one. The authors speculate that Arg-lipid interactions stabilize the helical kink, but could allow increased anion conductance during turnover.

Validity

The use of complementary techniques is impressive, and helps bolster the findings from each individually. The agreement with the previous characterization of GltTk lends further confidence to the results. I do feel that overall validity is harmed somewhat by some missing data in the current form (see below), for both biochemical and structural aspects. These can mostly be resolved by changing the visualizations.

Significance

This manuscript describes an impressive set of biochemical and structural investigations which demonstrate the value of GltTk as a model for a serious human disorder, and could potentially lead to clinically useful insights into EA6. The failure to identify obvious structural correlates of modified function in the P208R structure, although quite interesting in itself, limits this somewhat, as the proposed mechanism is essentially speculative and relies on MD and changes in unobserved states.

Specific comments about data and analysis

SSM experiments

The SSM experiments are in general adequately described and appropriately conducted, but some details and important data are missing. Although the authors cite Krause et al 2008, where mEAAC1 is extensively characterized similarly with SSM electrophysiology, the two proteins do have different transport mechanisms (GltTk lacks the H⁺ and K⁺ coupling of mEAAC1), and so there are still some questions about how differently they will behave.

1 - Experimental details for SSM perfusion should be expanded. The flow rates, times for each perfusion step, QC ranges for capacitance and conductance, and details about how many times each sensor was used aren't clearly indicated.

We fully agree with this reviewer that the experimental setup should be more thoroughly described, hence we have extended the materials and methods section with "solid-supported membrane experiments" section

2 - The normalization process in panel 2 B should be explained.

We have added the explanation for this in the figure legend

3 - Full traces for the double exchange experiments should be shown somewhere, even if just in the SI. The representative traces in panel 2 A appear to show the NA and A phases but not the perfusion with R buffer. Also, how many were performed?

Representative of full traces of Figure 2 are now shown in Suppl. Figure 5. The Figure 2 has been also modified

In a double solution exchange protocol, a third solution is included (resting solution, R) containing the same composition as the inside of the liposomes. This solution allows the generation of a gradient of a co-ion (or co-substrate) previous to the perfusion of activating solution (A) containing the substrate. This solution is perfused after every NA-A-NA exchange, such that the composition inside the liposomes is kept constant in between experiments. Regardless of the type of experiment (single solution or double solution exchange) the software configuration in the SURFE2R N1 only allows the current detection during the change NA-A-NA, therefore, the electrical signal from the NA-R exchange is not detected. Thus, the measurement window of the SSE and DSE experiments will show only the electrical information from the NA-A-NA solution. The electrical signal from the R-

NA exchange might appear as a transient current, but it is usually overlapped with the electrical artifact from the starting of the internal valves and, therefore, its interpretation is limited.

4 - Full traces for the empty liposome experiments should be included. Also, it appears that portions of the +L-Asp trace in SI Figure 2 B and the 300 mM NaSCN trace in SI 2 C are cut out - these should be shown.

This information has been added to the Supplementary Figure 4

5 - What are the samples and experimental conditions in SI Figure 2 A?

This information has been added (Now it is SI Figure 7)

6 - What concentration was the NaSCN jump in SI Figure 2 B? What concentration was L-Asp?

This information has been added (Now it is SI Figure 7)

Data fitting

The KM values are fit to data while allowing the Hill coefficients to vary freely between mutant and WT. The logic behind this should be more explicit (should cooperativity vary between them?), or some sensitivity analysis should be performed (do KM values change if n is fixed?).

For symporters, the apparent Hill coefficient n (where "apparent" indicates that the value depends on concentrations of the substrate and co-transported ion) gives insights in the kinetic mechanism of transporters as reported in (Lolkema and Slotboom, 2019) and (Trinco *et al.*, 2021). Specifically, it has been reported that for GlT_{TK}, the n values change based on the substrate concentration. A priori, we did not want to exclude the possibility that the two protein variants might have an altered kinetic mechanism, and therefore we allowed the Hill coefficient to vary in the regression analysis. It turned out the Hill coefficients of the wild-type and mutant proteins did not differ significantly.

Data interpretation

The claims about the effects of P208R on anionic conductance are slightly confusing to me. Line 343 says the "currents generated by NaSCN on GlT_{TK}_P208R were not significantly different from the ones in the wild type." The Figure 3 legend says "P208R mutation...increases the anion current amplitude." The relative amplitudes are not compared as far as I can see, as Figure 2 B compares normalized currents and finds the Na-dependence is indistinguishable. This should be clarified.

We apologize for the mistake. The statement that anionic conductance is not changing much was incorrect. We had not explicitly taken reconstitution efficiencies of the replicate experiments into account in the analysis of fig 3. We now have carefully quantified the amounts of wild-type and mutant protein used in the experiments of figure 3 (see SI figure 3), and now we can confidently conclude that indeed the P208R mutation increases the anion current amplitude.

L348: I am not sure one can infer anything about relative open probabilities from these results. I am convinced that the relative anion to L-Asp conductance is increased in P208R, but the results as shown seem to imply diminished L-Asp transport and identical anion conductance. A different experiment would be necessary to make a claim about P(open).

The reviewer is correct. As mentioned in the previous point, we now have carefully quantified the amounts of wild-type and mutant protein used in the experiments of figure 3 (see new SI figure 3), and now we can confidently conclude that the P208R mutation increases the anion current amplitude. We have rephrased the text accordingly.

Cryo-EM structure determination

I am not a cryo-EM expert, but I believe some necessary details are missing from the text and SI.

1 - The local resolution needs to be shown, not just the FSC map.

This information has been added to the SI figure 2

2 - The orientation preferences of particles should be shown.

This information has been added to the SI figure 2

3 - Does the second class during 3D classification post-ab initio represent junk particles or alternative states?
We could not extract any high or mid-resolution class out of this class, hence the assumption is indeed that it mainly contains junk with perhaps some very low populated states

4 - Some details about grid type and freezing conditions would be nice, but not necessary.

This information has been added to the materials and methods section

Minor notes

Figures 2, 3 - The error bars for the SSM and radioactivity data are not defined. Are they SEM, SD, a confidence interval? How were the uncertainties calculated, as well?

The figure legends have been updated accordingly.

Figure 4 - What level are the densities being contoured at?

This information has been added to the figure legends

SI Figure 2 D - the color scheme here is a bit confusing, with the gray tones not following the concentration gradient.

We have modified this panel.

Structure refinement - Calculating the Rama Z-score would be useful, as 0 outliers could indicate overrefinement.

We fully agree with this reviewer, we followed the Nature group template, and this is the sole reason why this information was not included. We have added it now to the table and in case we will be asked to conform to the template we will move this information to the materials and methods section. The overall Z-score is 0.40, which indicates the high quality of our structure.

Suggestion for clarity

A cartoon representation of the experimental schemes for the SSM experiments could help the reader follow them in Figures 2 and 3.

We have included it in SI figure 4.

A cartoon for the proposed scheme in Figure 4 would also help the reader integrate the MD results with the overall functional data.

We have modified Figure 4

References

L330: A citation for the claim about reduced transport rates in mammalian homologs would be appropriate.

We apologize for this oversight and now we have added the reference (Divito *et al.*, 2017)

Reviewer #2 (Remarks to the Author)

Review Guskov Manuscript

Members of the EAAT family of transporters play critical roles in regulating the concentration of the neurotransmitter glutamate in the mammalian central synapse. In addition, mutations in this transporter have been shown to cause a rare genetic disease called episodic ataxia type 6 (EA6) primarily affecting the cerebellum. Previous work has examined the functional effects of this mutation on mammalian EAAT family members, but the structural and mechanistic consequences of the mutation have yet to be understood.

Here, Colucci and co-authors set up a model system to study the EA6 mutation using a bacterial homolog of the EAATs, GltTk. They demonstrate that GltTk shows analogous functional changes due to the mutation to those in the EAATs, then determine the structure of the mutant protein and perform molecular dynamics simulations to probe the mechanism of the change.

This paper is essentially divided into two parts; in the first the authors nicely establish assays for the transport activity and anion conductance of GltTk using solid-supported membrane (SSM) measurements. The assays reported here are robust and the data strongly support the idea that the GltTk mutation P208R has parallel effects to the EA6-causing mutation in EAAT1. In the second part of the paper the authors solve the structure of GltTk carrying the P208R mutation and perform some molecular dynamics calculations. Unfortunately, since the

structure is solved in an outward-facing conformation which is not the anion-conducting state, these results fail to yield insight into the mechanism of the changes caused by the mutation and do not support the claim in the abstract that the salt bridge between the introduced arginine and lipid is “responsible for the widening of the anionic channel.”

In principle, the functional assays reported here would be of interest to a specialty journal audience, but the mechanistic claims are completely unsupported by evidence.

Major Concerns:

1. The structural and molecular dynamics data do not support the conclusion that the anion channel is widened to cause the increase in anion conductance.

Recent reports from the Ryan lab clearly demonstrate the EAAT anion channel is formed when the transporter is in an intermediate state in the transport process close to the inward-facing state and quite different from the outward-facing state solved here. Indeed, there is no evidence of an anion channel in the original outward-facing structures of GltPh. The authors here make no attempt to predict the equivalent state of the GltTk protein; indeed, their only comment about the anion pore in the structure is the statement “In principle these interactions could lead to a change in the channel geometry during turnover.” Thus, their claim in the abstract about “widening of the anionic channel” is without any experimental basis. Similarly, though the simulations seem to support the idea that the protein-lipid salt bridge is stable, they do not provide insight into the nature of the anion pore.

We apologize for the confusion caused with ambiguous phrasing, which we have corrected in the revision. We never wanted to claim that we have solved the anion-conductive state. The work from the Ryan lab on GltPh indeed revealed the conductive state associated with in the intermediate position of the elevator domain which was possible to arrest only via cross-linking. We did not have such an aim to obtain the similar state but rather to perform functional characterization of the anion conductance for mutant P208R and to resolve the structure of P208R at sufficient resolution to unambiguously describe the position of a mutated residue. But we realize that it has caused confusion. Given the similarity between GltPh and GltTk (77% sequence identity), we now included a model of GltTk based on GltPh in the anion conducting state model (PDB 6WYK) and recalculated the stability of a salt bridge between R208 and the phospholipid head group. Based on this we rephrased the conclusion, stating that the most plausible explanation of an effect of a mutant is that the transport domain lingers in a state compatible with anion conduction state longer compared to the WT, which leads to the reduced substrate transport. The situation might still differ in EAAT1, but this is beyond the current work.

2. The authors argue that the anion conductance is increased in P208R; however, their SSM data contradict this claim. Although they show that sodium and L-aspartate are linked to anion conductance in the wild type, the mutation does not seem to impact it.

Once again we apologize for the confusion – see our reply to reviewer#1

Minor:

1. Figure 1B is introduced on page 2 (84) for the first time without referring to residue P208. The reader is confused about the residue number since only P290 of EAAT1 is introduced.

We have changed the Figure 1 legend

2. In some figures, the structural representation should be improved to create a better visual aid for the reader. For example, figure 1B could be more prominent and less transparent. In figure 4C, please clarify the orientation shown. The zoom in figure 4A obstructs the structure.

We have changed the figures to improve visibility

3. In the SSM current traces in figure 2, there seems to be a clear time difference between the peak of L-aspartate vs. anion-induced currents. It would be nice if the authors discussed potential reasons for the shift.

For the L-Asp-induced currents, the peak shapes change with the Na⁺ concentration, which is caused by the kinetic mechanism of Na⁺ and Asp binding to the transporter. The first sodium ion binds with low affinity and low on-rate, followed by, respectively, the second Na⁺, Asp and the third Na⁺. At low sodium ion concentration, the dominating fraction of transporters is not occupied with Na⁺ at the onset of aspartate addition (because of the low affinity for Na⁺). In this situation, the cooperative binding of Na⁺ and Asp, and subsequent transport is detected, in which the binding of the first sodium ion is rate limiting. In contrast, at high sodium concentration, a large fraction of the transporters is already occupied with sodium ions at the moment that Asp is added, which leads to rapid binding of aspartate and the last sodium ion, and subsequent transport, which is evident from a different peak shape. For the SCN⁻-induced currents, the system is primed with Non-Activating buffer containing both Na⁺ and L-Asp. Since Na⁺ and L-Asp bind highly cooperatively, a high fraction of the transporters is fully occupied at the onset of SCN⁻ addition, at all Na⁺ concentrations used. Therefore SCN⁻ initially dissipates the positive-inside potential established by Na⁺-Asp symport in the NA condition, which leads to further Na⁺-Asp symport, as well as

the dominating SCN^- flux. Therefore, the time constants of this combined activity are indeed different from the L-Asp induced currents.

4. Between figures 2 and 3, the color scheme should be consistent for mutant and wild type, specifically wild type data in Fig.2B, compared to the rest. Also, it is not clear why the anion conductance data in Figure 2C are inverted relative to the equivalent data in the bottom of Figure 3B. Both should either be normalized as in Figure 3 or directly shown as in Figure 2.

We have modified the figure and clearly stated at each curve whether it is WT or P208R mutant.

5. Line 173 the molar ratios of GltTK:lipid:MSP is reported as 3:5:100. I suspect this is erroneous and should be either 3:100:5 or 5:100:3, as I don't expect that there are 100 MSPs per nanodisc with 5 lipids.

We have fixed this mistake

6. In several places, there are typos that could be improved (e.g., page 1(17), ... a reduction in the substrates substrate transport, ...)

The text was proof-read once again to avoid such typos.

7. The authors should be consistent with American vs. British spelling in text and figure (e.g., leveled vs. levelled or normalized vs. normalised)

The text has now been proof-read by a native speaker

Reviewer #4 (Remarks to the Author)

EAAT transporters, especially expressed in neural cells, have double function of Na^+ -L-Glu symport and Cl^- conductance. Pro to Arg mutation in the TM5 reduces Glu transporting activity while Cl^- efflux increased, which causes reduction in neural cell volume and apoptosis, associated with a disease called episodic ataxia 6 (EA6). This manuscript uses archaeal homologue of Asp transporter Glt(TK), belonging to the same SLC1A family to elucidate the functional mechanism of anion conductance and of EA6. First, the authors performed solid-state membrane electrophysiology to confirm that the archaeal homologue can be used as model of EAATs. Then, they confirmed mutation of Pro208 to Arg in Glt(TK) elevated SCN^- conductance, while Asp transporting activity was reduced. Next, the authors solved the Cryo-EM structure of Glt(TK)_P208R mutant at 3.25 Å overall resolution. The structure showed that the mutation remains the kink of TM5 as WT, but Arg208 provides tight electrostatic interaction with the phosphate head group of lipid. MD simulation revealed that the interaction causes lateral movement of TM5, leading to the widening of anion channel.

The functional and structural analyses are solid. However, this reviewer questions why the authors did not use human EAAT transporters for the functional and structural analyses, instead of the archaeal homologue, since the Cryo-EM structures of human EAATs have already been solved. This is a major concern by this reviewer.

This is a valid concern, but we believe that it is very exciting that the behavior of the P to R mutant is so well conserved in this family of transporters, which adds to the robustness of our findings. In addition, historically, our work was started well before the human structures were solved. Given the high degree of structural and functional conservation in this family of transporters, we can conclude that such a mutation has no impact on the global organisation of transmembrane helices.

In addition to this concerns, this reviewer raises the following concerns as follows.

1. In the abstract, the authors described archaeal homologue of glutamate transporters Glt(TK). This description is misleading, since Glt(TK) is an Asp transporter.

This is an accepted way that GltPh and GltTk are named archaeal homologues of glutamate transporters (for instance PMID: 23792560 and PMID: 35452090), but obviously both are aspartate transporters, this is mentioned in the text.

2. For widening of anion channel in MD simulation, did the authors not observe any structural change of the channel in Cryo-EM structure? Density of anion such as SCN^- could not be observed? At least more detailed description for anion channel widening by MD simulation should be described.

The cryo EM structure did not reveal the anion channel. This is a common problem, because it opens transiently, and can be captured to some extent only by mutagenesis and crosslinking. We have now added new MD simulations where we modeled the anion conductive state based on the structure of GltPh. We evaluate the stability of the salt bridge formed between the Arginine side chain (in place of a Pro residue) and a phospholipid in this state, and show that it is consistent with channel widening, but we prefer not to overinterpret it but rather again used it to estimate a salt bridge stability.

3. Comparison of the P2087R mutant with that of WT Glt(TK) did not reveal structural change such as lateral movement of TM5 ?

See the previous point

Reviewer #4 (Remarks to the Author):

The manuscript by Colucci et al. uses a homolog of human glutamate transporters, the archaeal GltTk to functionally and structurally characterize a proline to arginine mutation that is related to the episodic ataxia type 6 (EA6) neurological diseases. First, they use solid-supported membrane electrophysiology, Isothermal Titration Calorimetry (ITC), and radioactively labeled substrate uptake assays to prove that the GltTk proline to arginine (P208R) mutant is a good model of the human glutamate transporter mutated in EA6 disease. Then, they used single-particle cryoEM to solve the structure of the GltTk mutant. This structure reveals a similar kink of the helix at the position of the point mutation and the arginine side chain protruding toward the membrane bilayer, suggesting its interaction with the lipids headgroups. To prove this interaction, the authors performed a molecular dynamics simulation of the mutant and observed a persistent interaction between the arginine and the lipid headgroups.

The information provided in the manuscript could be of interest to the field of the glutamate transporter, but some issues need to be addressed to establish the validity of the presented conclusions. I will focus mainly on the computational results, which is my area of expertise while noting some issues from the experimental section.

1)- It is concluded that the ITC shows similar substrate binding affinities of the wild type and mutant, suggesting that the binding site does not change upon mutation. However, even if both dissociation constants are in the nanomolar range, the mutant KD measured in this work is ~11 nM while the wild type, measured also by ITC in a previous article, is 75 nM (at the same concentration of sodium), almost 7 fold difference. The measurements were performed in slightly different conditions, i.e. different buffers, and different instruments, thus it might be worth measuring again the wild-type KD to compare with the mutant values presented in this work.

This is a valid concern and it seems that indeed that different instruments behave somewhat differently – the affinity for L-Asp for WT using the same instrument as used here for P208R mutant gave a value of 36 nM. Furthermore, it might well be that the APPARENT affinity for L-Asp changes (even though the binding site remains identical in geometry) when the protein dwells for longer in a conformation that does not allow the opening of the gate (i.e. the chloride conducting state).

2)- Why did they perform the simulations using the wild type inward-inward-outward trimer? But also, it might not be the fairest comparison with their fully outward structure. Maybe simulations of the outward-occluded crystal structure or the TBOA fully outward cryo-EM structure are better controls for the mutant simulation.

See also response to reviewer #3. We have now carried out additional all-atom MD simulations of the anion-conductive state GltTk P208R model (derived from the threading) and this information has been added to the Supplementary Information (Supp. Fig. 12-14)

3)- From the RMSD progression, some protomers do not seem very stable in their simulation. Especially the one shown in the bottom pane of Fig S7 for the wild-type simulation. Does that correspond to the outward-facing protomer? Can you please discuss which kind of movements corresponds to the larger RMSD values?

- This information has been added to the Supplementary Fig. 12.

4)- In the simulations, the authors used the same lipid mixture (3 POPE: 3 POPG: 2 POPC) present in the nanodisc. However, lipid mixtures converge slowly (in the order of hundreds of nanoseconds). Hence, they should indicate how they placed lipids in the bilayer (i.e. are their positions randomized? How?). They also need to indicate whether the lipids are equilibrated or not.

-In this work, we used the CHARMM-GUI tool to build the membrane, which randomly distributes different types of lipids in accordance with the selected ratio.

5)- The simulations might need to be repeated from different starting points to establish the robustness of the observed interactions (maybe a total of three simulations per system).

- This has been done and now it is better described in the Supplementary Information

6)- Since the arginine-POPG interaction is not reversible in their simulation time scale, the authors should indicate whether the arginine in the P208R mutant binds to one specific POPG molecule or it exchanges freely with different POPG lipids in the simulation box.

-Yes, Arg exchanges freely with different lipid molecules.

7)- The authors speculate that this interaction between the arginine and the lipid headgroup might contribute to the widening of the chloride channel during the elevator movement. Do they observe more waters around the suggested pore in the simulations of the mutant, compared to the same conformational state of the wild-type?

- In brief – yes, this information has been added to the Supplementary Fig. 13 and Supplementary Movie 1.

Minor issues:

- Can you please discuss or speculate possible reasons why the mutant seems to stabilize the fully outward symmetric trimer? This conformation is not observed for the wild-type GlTk under a similar experimental setup, even in saturated aspartate conditions (Arkhipova et al., 2020 Nat Comm), so it might be a result of the P208R mutation.

This indeed might be an effect of the mutation, but we would like not to speculate too much about it. Another reason could be a slightly different ratio achieved, which could also facilitate the movement of transport domains

- Can you compare the newly reported structure with the TBOA fully outward protomers?

The structures are very much alike (rmsds ~ 0.9 Å)

- In the introduction section they mentioned that the ITC and the substrate uptake measurements showed reduced transport of the substrate, they should list only the uptake assay; the ITC showed only equilibrium binding.

ITC here relates to the binding affinity, not the transport

- 6XWO is a cryoEM structure, not a crystal structure, please correct that in the methods section.

Fixed

REVIEWERS' COMMENTS

Reviewer #1 (Remarks to the Author):

I thank the authors for their thorough, detailed responses to the reviewer comments. I appreciate the effort they expended in preparing this response and believe the the clarity and strength of the manuscript has been significantly improved in the process. I feel that all issues have been addressed and that the manuscript is definitely suitable for publication in this form.

Reviewer #2 (Remarks to the Author):

I commend the authors for their diligent efforts in revising the manuscript. Overall, the authors have robust functional and structural data that establish that GltTK is a valid model for examining the mechanism of mutations leading to human EA6. Though the authors have addressed almost all of my concerns, their statements regarding the possible mechanism by which the mutation alters the anion conductance are still without basis in data presented here.

The functional data shown in the manuscript nicely demonstrate the existence of an anion conductance in GltTK, and they further demonstrate alterations in the ratio of aspartate transport to Cl⁻ conductance. In the discussion, the authors appropriately discuss the possible alternative mechanisms for the change in anion conductance, but they still overstate their conclusion in several places. Most importantly, the statement in the abstract about the possible explanation of the mechanism (lines 26-30) should be removed, as none of the results directly support this idea. Also, the line about channel widening at the end of the introduction (line 109) should be removed. The conclusion stated at the end of the results (line 410) that the P208R mutation in the threaded model leads to pore widening is also not apparent from examination of the data presented in the supplement.

Reviewer #4 (Remarks to the Author):

My previous concerns were all addressed. I prefer the authors add a figure with the ITC data (the one that they measured a KD of 36nM) for the wild type (maybe a supplementary figure).

We are delighted to see that the reviewers are in general satisfied with the revised manuscript

Reviewer #1 (Remarks to the Author):

I thank the authors for their thorough, detailed responses to the reviewer comments. I appreciate the effort they expended in preparing this response and believe the the clarity and strength of the manuscript has been significantly improved in the process. I feel that all issues have been addressed and that the manuscript is definitely suitable for publication in this form.

We thank this reviewer for appreciation of our work

Reviewer #2 (Remarks to the Author):

I commend the authors for their diligent efforts in revising the manuscript. Overall, the authors have robust functional and structural data that establish that GlTK is a valid model for examining the mechanism of mutations leading to human EA6. Though the authors have addressed almost all of my concerns, their statements regarding the possible mechanism by which the mutation alters the anion conductance are still without basis in data presented here. The functional data shown in the manuscript nicely demonstrate the existence of an anion conductance in GlTK, and they further demonstrate alterations in the ratio of aspartate transport to Cl⁻ conductance. In the discussion, the authors appropriately discuss the possible alternative mechanisms for the change in anion conductance, but they still overstate their conclusion in several places. Most importantly, the statement in the abstract about the possible explanation of the mechanism (lines 26-30) should be removed, as none of the results directly support this idea. Also, the line about channel widening at the end of the introduction (line 109) should be removed. The conclusion stated at the end of the results (line 410) that the P208R mutation in the threaded model leads to pore widening is also not apparent from examination of the data presented in the supplement.

We have toned down our statements and clearly stated that only experiments on human EAAT1 in the anion conducting state will give the ultimate answer regarding what exactly is happening with the anion channel in EA6. Nevertheless, we are confident in our MD results which showed the increase of the calculated area of the water density map, which is indicative of the pore widening

Reviewer #4 (Remarks to the Author):

My previous concerns were all addressed. I prefer the authors add a figure with the ITC data (the one that they measured a KD of 36nM) for the wild type (maybe a supplementary figure).

We have remeasured Kd of a mutant (from the same batch which was used for electrophysiology experiments) with our preferred machine MicroCal PEAQ-ITC and it yielded the value of ~48 nM, which is very close to WT. Hence we decided to include this new value instead. The text has been modified accordingly.